# Porcine-Derived Chondroitin Sulfate Sodium Alleviates Osteoarthritis in HTB-94 Cells and MIA-Induced SD Rat Models

**DOI:** 10.3390/ijms26020521

**Published:** 2025-01-09

**Authors:** Hyelim Kim, Jinhee Kim, Seong-Hoo Park, Jinhak Kim, Yuri Gwon, Minhee Lee, Soo-Jeung Park

**Affiliations:** 1Research Institute of Clinical Nutrition, Kyung Hee University, Seoul 02447, Republic of Korea; hlimkim46@khu.ac.kr (H.K.); jinhee625@khu.ac.kr (J.K.); phoo3166@khu.ac.kr (S.-H.P.); 2R&D Division, Daehan Chemtech Co., Ltd., Seoul 01811, Republic of Korea; jhkim@dhchemtech.com (J.K.); rnd@dhchemtech.com (Y.G.); 3Department of Food Innovation and Health, Kyung Hee University, Yongin 17104, Republic of Korea; 4Beirne B. Carter Center for Immunology Research, University of Virginia, Charlottesville, VA 22903, USA

**Keywords:** osteoarthritis, chondroitin, joint cartilage, HTB-94, MIA, inflammation, apoptosis

## Abstract

Osteoarthritis (OA) is a chronic disease characterized by cartilage degradation, leading to bone friction, inflammation, stiffness, pain, and reduced mobility. This study investigates the therapeutic effects of porcine-derived chondroitin sulfate sodium (CS) on OA symptoms at both cellular and animal levels. *In vitro* study, HTB-94 chondrocytes were treated with inflammatory stimuli and CS (10, 50, 100, and 200 μg/mL) to assess the release of inflammatory mediators and the expression of genes and proteins related to cartilage synthesis and degradation. *In vivo* study, an MIA-induced OA rat model was used, and CS (62, 124, and 248 mg/kg b.w.) was orally administered for 4 weeks. Key parameters, such as exercise capacity, micro-CT, histological evaluation of joint tissues, serum inflammatory markers, and the expression of mRNA and proteins (inflammatory, cartilage synthesis and degradation, and apoptosis markers), were analyzed. Porcine-derived CS significantly reduced PGE_2_, NO, and extracellular matrix degradation marker (COMP and CTX-II) levels and increased the expression of cartilage synthesis-related genes and proteins in both HTB-94 cells and the MIA-induced rats. Additionally, CS modulated cartilage degradation pathways and notably inhibited apoptosis in vivo. The effects of porcine CS were comparable to the NSAID ibuprofen, demonstrating its potential as an anti-inflammatory and chondroprotective agent for OA management and dietary supplementation.

## 1. Introduction

Osteoarthritis (OA) is a degenerative joint disorder affecting over 650 million people globally, contributing to a substantial socioeconomic burden [1]. Recent studies have identified a range of individual and societal factors contributing to the development of OA, including occupation, sports participation, musculoskeletal injuries, obesity, and gender [2,3]. This condition is marked by progressive articular cartilage degradation, joint inflammation, and maladaptive subchondral bone remodeling, which collectively result in severe pain and chronic joint dysfunction [4]. Although various symptomatic treatments and physical therapies are utilized to alleviate patient discomfort, no disease-modifying interventions have been identified to address the underlying pathology of OA [5]. Preclinical research has explored several pharmacological strategies, including matrix-degrading enzyme inhibitors, anti-inflammatory molecules, and anabolic growth factors, which have shown potential in mitigating cartilage damage [6,7]. However, despite encouraging preclinical outcomes, many of these drug candidates have encountered significant challenges during clinical trials, such as insufficient efficacy and adverse side effects, ultimately hindering their advancement to clinical use [8]. One of the key obstacles in OA pharmacotherapy lies in maintaining effective drug concentrations within the vicinity of chondrocytes. Due to the avascular nature of articular cartilage, systemic drug administration—such as intravenous injections—often proves inadequate for achieving sufficient bioavailability in the joint. Consequently, addressing the delivery challenges inherent to OA remains a critical focus for developing future therapeutic solutions [9].

Articular cartilage is a smooth, resilient connective tissue that covers the ends of bones, playing a crucial role in reducing joint friction, ensuring smooth movement, and distributing mechanical loads efficiently across the joint surface [10]. When cartilage is damaged, the outermost superficial layer is disrupted, leading to the degradation of essential components of the extracellular matrix (ECM), including collagen fibers and proteoglycans (PGs) [11]. Such localized damage to chondrocytes (the resident cartilage cells) may progressively extend into the middle and deep zones of the tissue, impairing joint function. Due to the avascular nature of cartilage, its self-repair capacity is extremely limited, resulting in long-term structural deterioration and functional decline [12,13]. Cartilage damage is significantly influenced by the activation of matrix metalloproteinases (MMPs), apoptosis of chondrocytes, and the presence of reactive oxygen species (ROS). The stimulation of MMPs leads to an increased breakdown of extracellular matrix (ECM) components, particularly collagen and its associated elements, which contributes to the progressive deterioration of cartilage integrity, underscoring the multifactorial nature of cartilage degeneration [14].

Chondroitin sulfate (CS), a heteropolysaccharide classified as a glycosaminoglycan (GAG), is a naturally occurring component found in the ECM of various connective tissues. It plays a pivotal role in regulating cellular mechanisms by interacting with ECM proteins and influencing signaling pathways. Through these interactions, CS contributes to the maintenance of structural integrity and cellular communication within cartilage tissue [15]. Multiple clinical studies have reported that CS slows the progression of OA by modulating cartilage degradation and promoting joint health, highlighting its potential as a disease-modifying agent for OA management [16].

CS is primarily extracted from the cartilage of cattle, porcine ears, nasal septa, chickens, and marine animals, with bovine-derived CS being extensively utilized in both clinical and non-clinical research. Notably, bovine-derived CS has been reported to induce anti-inflammatory and anabolic pathways [17]. Most studies on chondroitin’s efficacy have focused on substances derived from cattle or sharks, leaving research on high-purity, 90% porcine-derived CS relatively limited. Consequently, the specific mechanisms and roles of porcine-derived CS, especially in osteoarthritis (OA), have not been thoroughly investigated [18].

This study aims to evaluate the non-clinical therapeutic effects and mechanisms of action of porcine-derived high-purity (90%) CS. To establish a cellular model, a human chondrosarcoma cell line will be utilized, and an OA experimental model will be designed using monosodium iodoacetate (MIA)-induced osteoarthritis in Sprague Dawley (SD) rats. Key assessments will include morphological evaluations, assessments of motor capacity impairment, changes in inflammatory factors, the generation and degradation of chondrocytes, and genetic evaluation of mechanisms related to chondrocyte apoptosis. Through these investigations, this research intends to provide a comprehensive overview of specific genes and biomarkers that are implicated in the symptom relief and therapeutic effects of porcine-derived CS in OA.

## 2. Results

### 2.1. Chondroitin Sulfate Sodium Alleviates PGE_2_ and NO Production in LPS-Treated HTB-94 Cells

Based on the results of the MTT assay performed on the HTB-94 cells treated with CS (0–1000 μg/mL), cell viability remained above 80% at concentrations up to 400 μg/mL. For subsequent experiments, CS was applied at concentrations ranging from 0 to 200 μg/mL, where no cytotoxic effects were observed (Figure 1A). We evaluated the effect of CS concentrations on the viability of HTB-94 cells treated with 200 μM H_2_O_2_. As the concentration of CS increased, cell viability improved correspondingly (Figure 1B).

The production of PGE_2_ was assessed in LPS-treated HTB-94 cells following treatment with varying concentrations of CS. The PGE_2_ level in the LPS-treated control group (C) was significantly higher compared to the NC group. However, the PC group exhibited a substantial reduction in PGE_2_ level relative to the C group. In the CS-treated groups (10, 50, 100, and 200 μg/mL), PGE_2_ levels decreased in a dose-dependent manner, with significant reductions observed sequentially in the CS 10, CS 50, CS 100, and CS 200 groups (*p* < 0.05; Figure 1C).

The NO production was evaluated in LPS-stimulated HTB-94 cells after treatment with different concentration of CS. The NO level in the LPS-treated C group was significantly elevated compared to in the NC group. However, the PC group exhibited a marked reduction in NO level relative to the C group. In the CS-treated groups (10, 50, 100, and 200 μg/mL), the NO levels decreased in a dose-dependent manner, with significant reductions observed progressively in the CS 10, CS 50, CS 100, and CS 200 groups (*p* < 0.05; Figure 1D). These findings established the experimental concentration of CS by optimizing its efficacy through in vitro experiments while ensuring its safety. Furthermore, it was demonstrated that CS effectively inhibits the production of PGE_2_ and NO, key mediators involved in inflammatory responses, in LPS-treated conditions at concentrations up to 200 μg/mL.

### 2.2. Chondroitin Sulfate Sodium Enhances the Expression of mRNA and Protein Associated with Cartilage Formation in H_2_O_2_-Treated HTB-94 Cells

The effects of CS on the expression of cartilage-regenerating genes and proteins were investigated in HTB-94 cells following treatment with H_2_O_2_. The mRNA expression levels of cartilage formation-related genes, including aggrecan, pro collagen type I, and pro collagen type II, were significantly downregulated in the H_2_O_2_-treated C group compared to those in untreated normal cells. Conversely, treatment with acetylsalicylic acid or CS following H_2_O_2_ exposure resulted in a significant upregulation of mRNA levels for aggrecan, pro-collagen type I, and pro-collagen type II compared to the H_2_O_2_-treated C group (*p* < 0.05; Figure 2A).

The expression levels of cartilage formation-related proteins, including aggrecan, collagen type I, and collagen type II, were markedly suppressed in the H_2_O_2_-treated C group compared to untreated normal cells. However, treatment with acetylsalicylic acid or CS after H_2_O_2_ exposure significantly restored the expression of these proteins. Notably, CS exhibited a dose-dependent effect, with increasing concentration (10, 50, 100, and 200 μg/mL) leading to progressively higher expression levels of aggrecan, collagen type I, and collagen type II compared to the H_2_O_2_-treated C group (*p* < 0.05; Figure 2B,C). These results demonstrate that CS plays a regulatory role in the expression of key factors involved in cartilage regeneration, such as aggrecan and collagen types. This was evidenced by comparative analyses of both gene and protein expression levels.

### 2.3. Chondroitin Sulfate Sodium Suppresses the Expression of mRNA and Protein Associated with Cartilage Degradation in H_2_O_2_-Treated HTB-94 Cells

The effects of CS on the expression of cartilage degradation-related genes and proteins were investigated in HTB-94 cells following treatment with H_2_O_2_. The mRNA and protein expression levels of cartilage degradation-related genes, including MMP-3 and MMP-13, were significantly increased in the H_2_O_2_-treated C group compared to those in untreated normal cells. Conversely, treatment with acetylsalicylic acid or CS after H_2_O_2_ exposure led to a significant downregulation of mRNA and protein expression levels of MMP-3 and MMP-13 in a dose-dependent manner, compared to the H_2_O_2_-treated C group (*p* < 0.05). Additionally, the expression levels of cartilage degradation-related mRNA and proteins, including TIMP-1, and TIMP-3, were significantly decreased in the H_2_O_2_-treated C group compared to those in untreated normal cells. However, treatment with acetylsalicylic acid or CS after H_2_O_2_ exposure lead to a significant upregulation of mRNA and protein expression levels of TIMP-1 and TIMP-3 in a dose-dependent manner compared to the H_2_O_2_-treated C group (*p* < 0.05; Figure 3). These findings demonstrate that CS contributes to the regulation of cartilage matrix homeostasis in vitro experiment by modulating the expression of MMP-3 and MMP-13, enzymes that specifically degrade type II collagen, and by enhancing the expression of TIMP-1 and TIMP-3, proteins that inhibit MMP activity. Both gene and protein expression analyses support these effects.

### 2.4. Oral Administration of Chondroitin Sulfate Sodium Mitigates Histology and Mineralization Impairments and Exercise Capacity in MIA-Induced Osteoarthritis Rat Model

This study found no significant differences in the body weight gain, food consumption, FER, and organ weight among all the groups (Table 1). Morphological evaluation of the knee joints in rats with MIA-induced osteoarthritis demonstrated pronounced pathological changes, including fibrillation, fissures, and extensive cartilage degradation (Figure 4A–C). Upon examining the macroscopic appearance of the tibia, the cartilage in the PC and CS groups exhibited only mild damage compared to the severe deterioration observed in the MIA-induced OA group (Figure 4A). Histological analysis using H&E staining revealed substantial improvements in fibrosis and fissure formation within the knee joint in the PC and CS groups, showing a near-normal condition compared to the pronounced pathological changes in the MIA-induced OA group (Figure 4B). Micro-Ct analysis showed that MIA injection significantly reduced BMD, BV/TV, Th.N, Tb.Th, and Tb.sp. However, treatment with ibuprofen or CS in MIA-injected rats mitigated these changes, improving bone quality in factors such as morphological changes and mineralization metrics. Specifically, CS treatment at doses of 62, 124, and 248 mg/kg b.w. significantly increased BMD, BV/TV, Th.N, Tb.Th, and Tb.sp in a dose-dependent manner (*p* < 0.05; Figure 4C,D).

Additionally, the MIA-injected rats exhibited significantly decreased pressure, propel, and running speed on the treadmill compared to normal rats, indicating that MIA induced impaired exercise capacity and osteoarthritis development. However, the oral administration of ibuprofen or CS treatment at doses of 62, 124, and 248 mg/kg b.w. ameliorated these impairments in a dose-dependent manner compared to MIA-injected rats that did not receive treatment (*p* < 0.05; Figure 4E). This study confirmed through morphological analysis that the intake of CS in animal models alleviated and regenerated the extensively degraded cartilage, fibrotic changes, and fissures in the joint regions. Furthermore, it was demonstrated that CS intake contributed to the improvement of exercise capacity, which had been diminished due to OA, thereby indicating its beneficial effects on restoring motor function.

### 2.5. Chondroitin Sulfate Sodium Ameilorates Cartilage Inflammation and Extracellular Matrix Degradation Markers in MIA-Induced Osteoarthritis Rat Model

The MIA-injected rats demonstrated significantly elevated serum levels of PGE_2_ and NO compared to normal SD rats, confirming the induction of inflammation. However, treatment with ibuprofen or CS at dose of 62, 124, and 248 mg/kg b.w. resulted in statistically significant, dose-dependent reductions in serum PGE_2_ and NO levels compared to MIA-injected control (*p* < 0.05; Figure 5A,B).

The MIA-injected rats demonstrated significantly elevated serum levels of COMP and CTX- II compared to normal SD rats, confirming the induction of extracellular matrix degradation markers. However, treatment with ibuprofen or CS at doses of 62, 124, and 248 mg/kg b.w. resulted in a marked reduction in serum COMP and CTX- II levels in a dose-dependent manner compared to MIA-injected control. Notably, the COMP marker exhibited the highest statistical significance at the high concentration of CS 248 mg/kg b.w. (*p* < 0.05; Figure 5C,D).

This result demonstrates that the intake of CS significantly attenuated the production of PGE2 and NO, which were induced during OA, thereby modulating inflammation-related pathways. Furthermore, CS intake was shown to regulate the levels of CTX-II, a peptide generated from the degradation of type II collagen, as well as the levels of COMP, a key mediator involved in the interaction between chondrocytes and ECM components. These findings indicate that CS modulates the structural damage and degradation of cartilage by regulating both the breakdown of type II collagen and the molecular interactions within the ECM, highlighting its potential to impact the progression of OA.

### 2.6. Chondroitin Sulfate Sodium Regulates the mRNA and Protein Expression Related to Cartilage Formation and Degradation in MIA-Injected Osteoarthritis Rat Model

The effects of CS on the expression of cartilage formation and degradation-related genes were investigated in a MIA-injected osteoarthritis rat model. The mRNA expression levels of cartilage formation-related genes, including aggrecan, pro collagen type I, and pro collagen type II, were significantly downregulated in the MIA-injected rats compared to the untreated normal rats. However, oral administration of ibuprofen or CS treatment at doses of 62, 124, and 248 mg/kg b.w. increased these mRNA expression levels in a dose-dependent manner compared to MIA-injected rats that did not receive treatment (*p* < 0.05; Figure 6A).

The mRNA expression levels of cartilage degradation-related genes, including MMP-3 and MMP-13, were significantly upregulated in the MIA-injected rats compared to the untreated normal rats. However, oral administration of ibuprofen or CS treatment at doses of 62, 124, and 248 mg/kg b.w. reduced these mRNA expression levels in a dose-dependent manner compared to MIA-injected rats. The mRNA expression levels of cartilage regulation-related genes, including TIMP-1 and TIMP-3, were significantly downregulated in the MIA-injected rats compared to the untreated normal rats. However, oral administration of ibuprofen or CS treatment at doses of 62, 124, and 248 mg/kg b.w. increased these mRNA expression levels in a dose-dependent manner compared to MIA-injected rats. (*p* < 0.05; Figure 6B).

The expression levels of proteins associated with cartilage degradation, including smad3, phospho-smad3, MMP-3, and MMP-13, were assessed. The ratio of phospho smad3/smad3 was reduced in the MIA-induced rats compared to the normal rats. Conversely, oral administration of ibuprofen or CS treatment at doses of 62, 124, and 248 mg/kg b.w. increased the ratio of phospho-smad3/smad3 in a dose-dependent manner compared to the MIA-injected rats. The expression levels of MMP-3 and MMP-13 proteins were upregulated in the MIA-induced rats compared to the normal rats. However, oral administration of ibuprofen or CS treatment at doses of 62, 124, and 248 mg/kg b.w. reduced these proteins in a dose-dependent manner compared to MIA-injected rats (*p* < 0.05; Figure 6C). This result provides evidence that the intake of CS regulates the expression of genes and proteins associated with cartilage formation and degeneration, thereby enhancing regenerative processes and inhibiting cartilage degradation. These findings demonstrate the functional role of CS in promoting cartilage repair and preventing the progression of cartilage breakdown in vivo OA model.

### 2.7. Chondroitin Sulfate Sodium Alleviates the mRNA and Protein Expression Related to Inflammatory Markers in MIA-Injected Osteoarthritis Rat Model

The effects of CS on the mRNA expression related to inflammatory markers were evaluated in MIA-injected osteoarthritis rat model. The mRNA expression levels of pro-inflammatory cytokines, including TNF-α, IL-1β, and IL-6, were significantly elevated in the MIA-injected rats compared to the untreated normal rats. However, oral administration of ibuprofen or CS at doses of 62, 124, and 248 mg/kg b.w. resulted in a dose-dependent downregulation of these mRNA levels compared to untreated MIA-injected rats (*p* < 0.05; Figure 7A).

The expression levels of proteins associated with inflammatory markers, including IκBα, phospho-IκBα, p65, phospho-p65, COX-2, and iNOS, were evaluated. The phospho-IκBα/IκBα ratio was increased in MIA-induced rats compared to the normal control. Oral administration of ibuprofen or CS at doses of 62, 124, and 248 mg/kg b.w. significantly downregulated the phospho-IκBα/IκBα and phospho-p65/p65 ratios in a dose-dependent manner compared to the untreated MIA-injected rats. Additionally, the expression levels of COX-2 and iNOS proteins were upregulated in the MIA-induced rats compared to the normal group. However, treatment with ibuprofen or CS at the aforementioned doses reduced these proteins in a dose-dependent manner compared to the MIA-injected control (*p* < 0.05; Figure 7B,C). This study demonstrates that CS intake regulates inflammation-related pathways and various genes and proteins mediated by p65, which were elevated in the MIA-induced rat OA model. By modulating these inflammatory processes, CS effectively reduced the inflammation levels in OA, highlighting its therapeutic potential for OA treatment.

### 2.8. Chondroitin Sulfate Sodium Alleviates the Protein Expression Associated with Apoptosis Mechanism in MIA-Injected Osteoarthritis Rat Model

The effects of CS on the protein expression related to apoptosis mechanisms were evaluated in MIA-injected osteoarthritis rat model. The expression levels of proteins associated with apoptosis mechanism, including JNK, phospho-JNK, c-Fos, phospho-c-Fos, c-Jun, phospho-c-Jun, FADD, caspase-8, cleaved caspase-8, bax, bcl-2, caspase-3 and, cleaved caspase-3, were evaluated. The ratios of phosphor-JNK/JNK, phospho-c-Fos/c-Fos, phospho-c-Jun/c-Jun, cleaved caspase-8/caspase-8, bax/bcl-2, and cleaved caspase-3/caspase-3 were increased in the MIA-induced rats compared to the normal control. Oral administration of ibuprofen or CS at doses of 62, 124, and 248 mg/kg b.w. significantly downregulated these ratios in a dose-dependent manner compared to the untreated MIA-injected rats. Additionally, the expression level of FADD protein was upregulated in MIA-induced rats compared to the normal group. However, treatment with ibuprofen or CS at the aforementioned doses reduced FADD protein in a dose-dependent manner compared to the MIA-injected control (*p* < 0.05; Figure 8A,B). This study reveals that CS intake regulates the gene and protein expression of factors involved in both inflammation-related mechanisms and apoptosis pathways in the MIA-induced rat OA model. Ultimately, CS modulates the activity of caspase-3, leading to the inhibition of apoptosis, thereby demonstrating its potential to mitigate cartilage cell death in OA.

## 3. Discussion

CS is found in the connective tissues of humans, other mammals, and invertebrates. It is particularly abundant in cartilage, skin, blood vessels, ligaments, and tendons, and is present in the ECM surrounding neurons and brain cells, where it serves as an essential component of PGs [19,20,21]. Orally administered CS alleviates pain not by increasing plasma concentrations but through its modulatory effects on cellular activities in the intestinal lining and liver, demonstrating efficacy comparable to NSAIDs. CS also aids in reducing the required dosage of NSAIDs in OA treatment. Unlike NSAIDs, CS exhibits prolonged residual effects that persist for several months, highlighting a distinct therapeutic advantage [22,23]. CS, a highly viscous mucopolysaccharide, appears slightly cloudy or pale yellow depending on its source, with lower-molecular-weight variants exhibiting potentially better absorption than high-molecular-weight forms. The origin and structure of CS significantly influence its functionality, quality, efficacy, yield, and identity. Once absorbed, CS accumulates at high concentrations in synovial fluid and cartilage tissue. It is well-tolerated, with no significant drug interactions or adverse effects reported, even at higher doses. Recent studies have reported that a two-step enzymatic treatment method employing alkalase and flavorzyme in the processing of CS not only reduces processing time but also enhances yield. These findings provide evidence that the chondrogenic and anti-degradative effects of CS contribute to the maintenance of cartilage health [17].

Articular cartilage exhibits a structure characterized by a high density of sulfate groups and negative charges, enabling it to bind water and cations (such as Na^+^), which generates electrostatic repulsion. This mechanism contributes to the cartilage’s resistance to compression and elasticity. CS enhances the synthesis of type II collagen and PGs, thereby promoting anabolic metabolism within the cartilage, while also exhibiting anti-inflammatory effects, stimulating hyaluronic acid production, and inhibiting the catabolic processes of chondrocytes. The loss of CS from cartilage is associated with the development of osteochondral angiogenesis, a recognized primary factor in OA progression [24,25]. Joint damage results in the breakdown of the ECM of cartilage, triggering the production of additional inflammatory mediators, such as TNF-α, IL-1β, and MMPs, including MMP1 and MMP3 [26]. In particular, levels of IL-6 increase in synovial fluid and serum, while cyclooxygenase-2 (COX-2) promotes the production of (PGE_2_) [27]. Joint inflammation activates damage-associated molecular patterns (DAMPs), leading to the recruitment of immune cells and the release of inflammatory mediators. Additionally, increased vascular permeability results in edema, contributing to elevated intra-articular pressure. Prolonged joint load can ultimately cause structural damage to the joint, resulting in pain and a decline in functional mobility [28]. Ultimately, these events not only contribute to cartilage degradation but also activate innate immune responses, perpetuating a cycle of tissue destruction and inflammation [29].

In this study, the HTB-94 cell line was utilized as a cartilage injury model, demonstrating that stimulation with either LPS or H_2_O_2_ led to a significant increase in the production of PGE_2_ and NO. Additionally, the expression of genes and proteins associated with aggrecan and collagen was downregulated, while genes involved in cartilage degradation were upregulated. The findings from this cell model align with recent studies using the HTB-94 cell line. Moreover, we utilized an established animal model using SD rats with MIA injection, which significantly elevated serum levels of PGE_2_ and NO. Additionally, we observed a downregulation of cartilage formation-associated factors and an upregulation of inflammatory cytokines, MMPs, cartilage degradation markers, and apoptosis-related pathways. These findings underscore the effectiveness of the MIA injection model as a suitable method for inducing OA, highlighting the significant inflammatory responses and cartilage damage observed in the experimental animals [30,31].

This study evaluated the effects of porcine-derived CS in a cartilage damage model. The results indicate that CS alleviated the levels of PGE2 and NO in the stimulated HTB-94 cell model and enhanced the expression of key factors related to cartilage formation, including aggrecan and collagen-associated proteins. Notably, CS significantly reduced the levels of extracellular matrix degradation markers COMP and CTX-II in a rat model of osteoarthritis. Furthermore, CS modulated the expression of matrix MMPs and TIMPs. The effects of porcine-derived CS were also pronounced in animal models, where treatment with CS resulted in a reduction in inflammatory markers at both serum and tissue gene expression levels, promoted cartilage regeneration, and improved locomotor function. Additionally, CS treatment decreased the expression of caspase-8 and caspase-3, thereby regulating apoptosis-related factors. These findings suggest that CS inhibits inflammatory and apoptotic signaling in cartilage, protecting the ECM and alleviating symptoms of OA, ultimately enhancing joint functionality.

Porcine-derived CS exhibits effects similar to those of non-steroidal anti-inflammatory drugs (NSAIDs) and is known for its minimal side effects even with long-term use, underscoring its potential as a substitute or dietary supplement. Until recently, the commercial supply and research involving CS have predominantly relied on raw materials derived from shark cartilage and bovine sources, with numerous studies reporting on their efficacy related to joint health. Recent studies on bovine-derived CS have demonstrated its efficacy in alleviating knee joint swelling and improving OA symptoms. Notably, micro-CT, a commonly used method for clinically diagnosing and observing OA, revealed a significant reduction in joint surface depression and an improvement in bone joint deformation following treatment with CS [16,32]. In addition, a study evaluating the efficacy of fish-derived CS on cell death in post-traumatic OA (PTOA) confirmed that it reduces the activation of NF-κB and p38 MAPK, thereby alleviating systemic inflammation and decreasing joint inflammation and cartilage degradation [33].

In this study, the efficacy of high-purity porcine-derived CS was evaluated, distinguishing it from previous studies on bovine- and fish-derived CS by addressing a broad range of factors, including inflammatory mechanisms, cell death, joint formation and regeneration, and degradation processes. This study not only compares the differences in gene and protein expression but also assesses physical performance, demonstrating that the therapeutic effects of high-purity CS are significantly enhanced. Additionally, advanced techniques have been developed for intra-articular mucosal supplementation using hybrid cooperative complexes of hyaluronic acid and CS. However, these improvements are often reported to be gradual rather than immediate [34]. In contrast, there is a lack of efficacy studies on porcine-derived chondroitin sulfate with a purity of 90%, highlighting significant research value in this area. Porcine-derived CS offers the advantage of high purity, which is a critical factor in maximizing therapeutic efficacy. High-purity CS contains fewer impurities, thereby minimizing potential side effects. Additionally, the cartilage structure of pigs is highly similar to that of humans, allowing for a more effective match with human cartilage, further enhancing the therapeutic outcome [16]. This results in not only high-efficiency therapeutic effects but also significant industrial applicability, as porcine-derived CS is amenable to commercialization and large-scale production, enabling the provision of therapeutic benefits to a broader patient population.

This study utilized high-purity (90%) porcine-derived CS to investigate its effects on cartilage generation and the inhibition of degradation factors, as well as its associated functionalities related to cartilage health. OA is a chronic condition that increases with age, with an estimated 9.6% of men and 18% of women aged 60 and above experiencing symptomatic OA worldwide [35]. Age is therefore considered a critical factor in the progression of the disease. However, this study focuses on preclinical research and has limitations regarding the age range of the sample used.

Future research could explore the potential long-term effects of oral administration, compare mucosal supplementation techniques, investigate the mechanisms of immune cells interacting with DAMPs, and examine the pathways of neuropeptides and cytokines acting as inflammatory mediators. Additionally, based on the results of these non-clinical studies, further clinical research on porcine-derived high-purity CS (90%) is warranted. Through such studies, new therapeutic strategies for OA may be formulated. We propose that the results of these investigations could pave the way for the industrial application and medical utilization of porcine-derived CS in promoting joint cartilage health.

## 4. Materials and Methods

### 4.1. Preparation of Chondroitin Sulfate Sodium (CS)

The CS (ScanDroitin^TM^) manufactured by ZPD A/S (Esbjerg, Denmark) used in the main experiment was provided by Daehan Chemtech Co., Ltd. (Gwacheon, Republic of Korea). The crude CS was derived enzymatically, hydrolyzed from porcine cartilage. The purification, filtration, concentration, and freeze-drying processes were performed to produce a product containing purified chondroitin sulfate sodium at a concentration of no less than 90%, calculated on a dry basis.

### 4.2. Cell Culture and Treatments

The human chondrosarcoma cell line SW1353 (HTB-94) used in this study was purchased from the American Type Culture Collection (Rockville, MD, USA). The cells were incubated and cultured in a complete high-glucose Dulbecco’s modified eagle’s medium (DMEM) supplemented with 10% FBS and 1% penicillin–streptomycin. The experimental procedure was adapted from the method described by Bergstrom A.R. [36].

The cells were seeded at a density of 2 × 10^5^ cells/well into a 6-well plate. Stabilized cells, excluding the normal control (NC) group, were treated with or without H_2_O_2_ (200 μM, 4 h) or LPS (50 μg/mL, 24 h), along with various concentrations of CS (10, 50, 100, and 200 μg/mL) or the positive control (PC), acetylsalicylic acid (10 μM).

### 4.3. Animal Study Design

SD rats (a total of 48 male, 6–8 weeks old) were purchased from Saeronbio, Inc. (Uiwang, Republic of Korea). The SD rats were housed in climate-controlled quarters at 23 ± 2 °C and 55% humidity (12 h light/dark cycle) [37]. The animals were acclimated for one week and group allocation was randomly assigned to six groups as follows: NC; MIA-induced control (C); positive control (PC; MIA + 20 mg/kg/day of ibuprofen); MIA + 62 mg/kg of CS (CS 62); MIA + 124 mg/kg of CS (CS 124); and MIA + 248 mg/kg of CS (CS 248). Osteoarthritis was induced by a single intra-articular injection of 50 μL of MIA solution (60 mg/mL) into the knee joint. Control rats were administered an equivalent volume of saline via the same route of injection. Oral treatments were administered by gavage once per day for 31 consecutive days. The order of treatments was randomized, and measurements were taken at consistent time intervals across all groups.

The Institutional Animal Care and Use Committee of Kyung Hee University approved the protocol (KHGASP-23-540) for the animal studies. The animals were maintained in accordance with the university’s Guidelines for Animal Experiments.

### 4.4. ELISA

The levels of prostaglandin E_2_ (PGE_2_) and nitric oxide (NO) in the supernatant of HTB-94 cells were quantified using ELISA kits (R&D Systems, Minneapolis, MN, USA). At the end of the animal experiment, blood samples were collected from the SD rats. Serum levels of PGE2, NO, COMP (Thermo Scientific, Rockford, IL, USA), and CTX-II (LSBio, Shirley, MA, USA) were measured, following the manufacturer’s protocol.

### 4.5. Real-Time Polymerase Chain Reaction (RT-PCR)

The mRNA was extracted from the treated HTB-94 cells and cartilage tissues isolated from the knee joints. The mRNA expression levels of aggrecan, pro-collagen type I, pro-collagen type II, MMP-3, MMP-13, tissue inhibitors of metalloproteinase (TIMP)-1, TIMP-3, TNF-α, IL-1β, IL-6, and GAPDH were quantified using real-time PCR, following previously established protocols [38].

### 4.6. Protein Extraction and Western Blot Analysis

The treated HTB-94 cells and cartilage tissues isolated from the knee joints were used for protein extraction. Western blot analysis was performed to determine the expression levels of aggrecan, collagen type I, collagen type II, MMP-3, MMP-13, TIMP-1, TIMP-3, Smad3, phospho-Smad3, IκB-α, phospho-IκB-α, p65, phospho-p65, COX-2, iNOS, JNK, phospho-JNK, c-Fos, phospho-c-Fos, c-Jun, phospho-c-Jun, FADD, caspase-8, cleaved caspase-8, Bax, Bcl-2, caspase-3, cleaved caspase-3, and actin, following previously established methods [38].

### 4.7. Evaluation of Exercise Capacity

Exercise capacity was assessed through a treadmill performance test (Jeollanamdo Institute of Natural Resources Research, Naju, Republic of Korea). Measured parameters included rear pressure, rear propulsion, and running speed. Prior to testing, each rat underwent acclimatization to the treadmill environment, and their exercise performance was systematically recorded.

### 4.8. Micro-Computed Tomography

Bone mineral density (BMD), bone volume to total tissue volume ratio (BV/TV), trabecular number (Tb.N), trabecular thickness (Tb.Th), and trabecular separation (Tb.Sp) were quantified using a micro-CT scanner (Skyscan 1176, Bruker, Belgium). High-resolution scans of the knee joints were performed, and the acquired images were processed and analyzed using dedicated micro-CT software CTAn (version 1.18, Bruker, Belgium).

### 4.9. Histological Observation

At the conclusion of the oral administration period, the rats were euthanized, and knee joints were collected for histological analysis. The harvested joints were initially examined through gross inspection to assess any visible abnormalities or morphological changes. The harvested joints were fixed in 10% neutral-buffered formalin, decalcified, and embedded in paraffin. Tissue sections (5 µm) were prepared and stained with hematoxylin and eosin (H&E) to evaluate morphology and assess cartilage degradation.

### 4.10. Statistical Analysis

Data analysis was conducted without any knowledge of group allocation, as the allocation information was removed prior to analysis, ensuring that the analysts were blinded to the groupings. All data are presented as the mean ± standard deviation (SD). Statistical analyses were conducted using one-way analysis of variance (ANOVA), followed by Duncan’s multiple-range test for post hoc comparisons to identify significant differences between groups. The SPSS statistical software (SPSS PASW Statistics 22.0, SPSS, Inc., Chicago, IL, USA) was used, and a *p*-value of <0.05 was regarded as the threshold for statistical significance.

## 5. Conclusions

This study demonstrates that the oral administration of porcine-derived high-purity (90%) CS significantly alleviated OA symptoms based on inflammatory indicators in both the HTB-94 cell line and the OA-induced animal model. Notably, porcine-derived CS effectively modulated factors associated with cartilage regeneration and degradation at both cellular and animal levels, while also inhibiting apoptosis. Furthermore, it improved the abnormal cartilage morphology, bone density, and locomotor abilities of rats subjected to MIA injections. These findings suggest that porcine-derived CS possesses significant value as a material capable of protecting chondrocytes, preventing the degradation of the ECM, and regulating inflammatory responses in vitro and in vivo research. The innovative aspect of porcine-derived CS lies in its high purity and quality, which minimizes impurities and reduces the potential for side effects. This makes it a promising functional food ingredient capable of effectively supporting cartilage protection and regeneration. It is already recognized as a safe material for long-term use, with its therapeutic effects being faster and more potent compared to bovine or fish-derived CS. Additionally, its ease of mass production enhances its potential for industrial applications, positioning it as a highly efficient material. Future research will focus on further investigation of molecular interactions and immune cell responses to elucidate the intricate signaling mechanisms involved. Clinical trials will also be conducted to establish the safety and efficacy of porcine-derived CS.

In conclusion, porcine-derived high-purity CS represents a promising candidate for enhancing current therapeutic approaches to arthritis, optimizing management protocols, and advancing the formulation of dietary supplements. The notable anti-inflammatory, anti-apoptotic, and protective properties of chondrocytes underscore their potential utility in these domains.

## Figures and Tables

**Figure 1 ijms-26-00521-f001:**
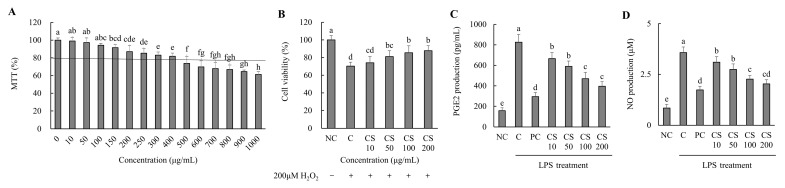
Effect of porcine-derived chondroitin sulfate sodium (CS) on the cell viability, cell protection, and levels of PGE_2_ and NO in H_2_O_2_ or LPS-treated HTB-94 cells. (**A**) MTT assay—the cells were incubated for 24 h with various concentrations of CS; (**B**) cell-protective ability—the cells were treated with/without 200 µM H_2_O_2_ and incubated for 4 h with various concentrations of CS; (**C**) PGE2 level; (**D**) NO level—the cells were treated with 50 µg/mL LPS, except NC, and incubated for 24 h with 10 µM of PC (acetylsalicylic acid) and various concentrations (10, 50, 100, 200 µg/mL) of CS. PGE_2_ and NO levels were analyzed by ELISA. The results are expressed as mean ± SD values (*n* = 3). Duncan’s test is used, and the different superscript letters (alphabet) indicate significance at *p* < 0.05.

**Figure 2 ijms-26-00521-f002:**
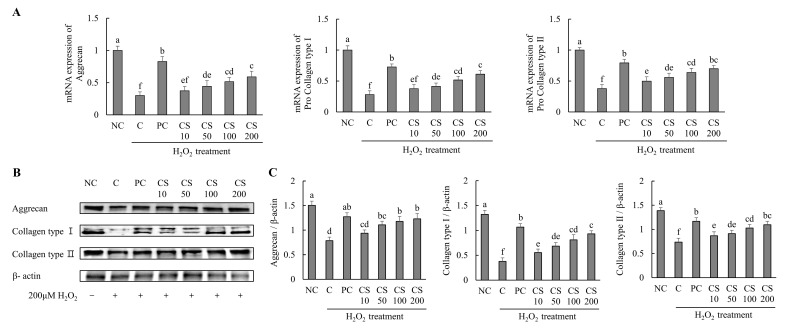
Effect of porcine-derived chondroitin sulfate sodium (CS) on the expression of mRNA and protein associated with cartilage formation in H_2_O_2_-treated HTB-94 cells. (**A**) The mRNA expression of aggrecan, pro collagen type I, and pro collagen type II. (**B**) Western blot analysis of cartilage formation-related protein expression. (**C**) Quantification of aggrecan/β-actin, collagen type I/β-actin, and collagen type II/β-actin. The cells treated for 24 h with 10 µM of PC (acetylsalicylic acid) and various concentrations (10, 50, 100, 200 µg/mL) of CS were incubated for 2 h with 200 µM H_2_O_2_. The mRNAs were analyzed by real-time PCR. The proteins were analyzed by Western blot. The results are expressed as mean ± SD values (*n* = 3). Duncan’s test is used, and the different superscript letters (alphabet) indicate significance at *p* < 0.05.

**Figure 3 ijms-26-00521-f003:**
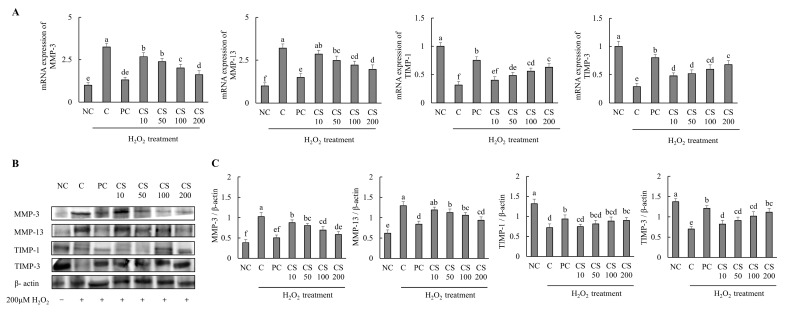
Effect of porcine-derived chondroitin sulfate sodium (CS) on mRNA and protein associated with cartilage degradation in H_2_O_2_-treated HTB-94 cells. (**A**) mRNA expression of MMP-3, MMP-13, TIMP-1, and TIMP-3. (**B**) Western blot analysis of cartilage degradation-related protein expression. (**C**) Quantification of MMP-3/β-actin, MMP-13/β-actin, TIMP-1/β-actin, and TIMP-3/β-actin. The cells treated for 24 h with 10 µM of PC (acetylsalicylic acid) and various concentrations (10, 50, 100, 200 µg/mL) of CS were incubated for 2 h with 200 µM H_2_O_2_. The mRNAs were analyzed by real-time PCR. The proteins were analyzed by Western blot. The results are expressed as mean ± SD values (*n* = 3). Duncan’s test is used, and the different superscript letters (alphabet) indicate significance at *p* < 0.05.

**Figure 4 ijms-26-00521-f004:**
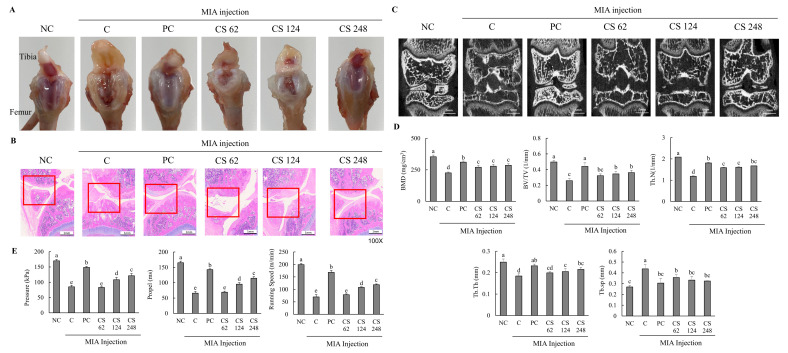
Effect of porcine-derived chondroitin sulfate sodium (CS) on macroscopic appearance of tibia, histological change and morphological features of hind knee joint, and exercise capacity in MIA-injected rat model. (**A**) Macroscopic appearance of tibia. (**B**) Histological change in hind knee joint by H&E staining. (**C**) Expression of coronal sagittal micro-CT image. (**D**) Morphological features of hind knee joint. (**E**) Measurement of exercise capacity by treadmill. NC, AIN-93G; C, AIN-93G + MIA injection; PC, ibuprofen 20 mg/kg b.w. + MIA injection; CS 62, CS 62 mg/kg b.w. + MIA injection; CS 124, CS 124 mg/kg b.w. + MIA injection; CS 248, CS 248 mg/kg b.w. + MIA injection. The results are expressed as mean ± SD values (*n* = 8). Duncan’s test is used, and the different superscript letters (alphabet) indicate significance at *p* < 0.05.

**Figure 5 ijms-26-00521-f005:**
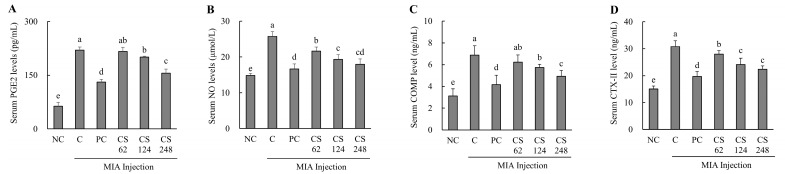
Effect of porcine-derived chondroitin sulfate sodium (CS) on cartilage inflammation and extracellular matrix degradation markers of serum in MIA-injected rat model. (**A**) PGE2 level; (**B**) NO level; (**C**) COMP level; (**D**) CTX-II level. NC, AIN-93G; C, AIN-93G + MIA injection; PC, Ibuprofen 20 mg/kg b.w. + MIA injection; CS 62, CS 62 mg/kg b.w. + MIA injection; CS 124, CS 124 mg/kg b.w. + MIA injection; CS 248, CS 248 mg/kg b.w. + MIA injection. The results are expressed as mean ± SD values (*n* = 8). Duncan’s test is used, and the different superscript letters (alphabet) indicate significance at *p* < 0.05.

**Figure 6 ijms-26-00521-f006:**
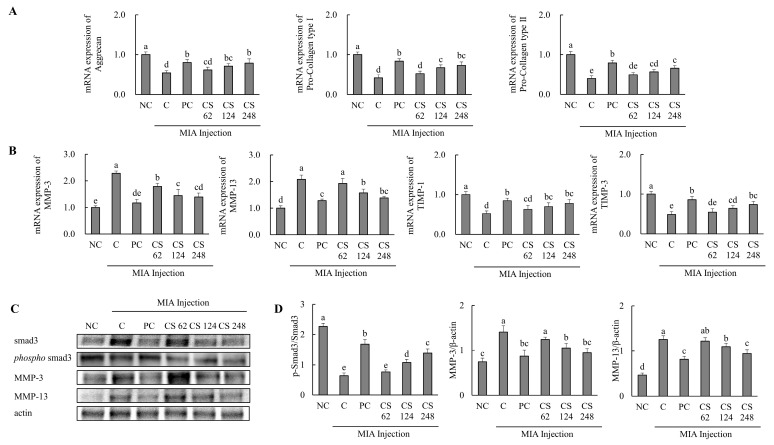
Effect of porcine-derived chondroitin sulfate sodium (CS) on the expression of mRNA and protein associated with cartilage formation or degradation in MIA-injected rat model. (**A**) The mRNA expression associated with cartilage formation (aggrecan, pro collagen type I, and pro collagen type II); (**B**) the protein expression related to cartilage degradation (MMP-3, MMP-13, TIMP-1, and TIMP-3); (**C**) Western blot analysis of cartilage degradation-related protein expression; (**D**) quantification of phospho-smad3/smad3, MMP-3/β-actin, and MMP-13/β-actin. NC, AIN-93G; C, AIN-93G + MIA injection; PC, ibuprofen 20 mg/kg b.w. + MIA injection; CS 62, CS 62 mg/kg b.w. + MIA injection; CS 124, CS 124 mg/kg b.w. + MIA injection; CS 248, CS 248 mg/kg b.w. + MIA injection. The results are expressed as mean ± SD values (*n* = 8). Duncan’s test is used, and the different superscript letters (alphabet) indicate significance at *p* < 0.05.

**Figure 7 ijms-26-00521-f007:**
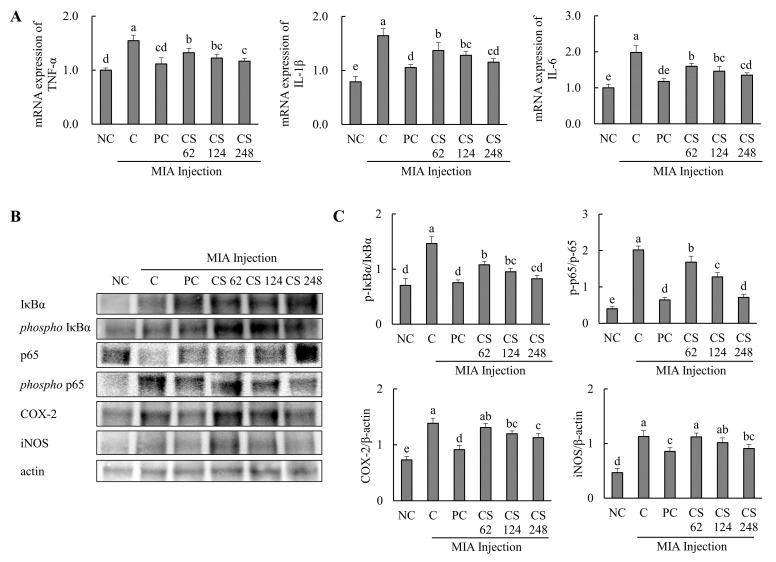
Effect of porcine-derived chondroitin sulfate sodium (CS) on the expression of mRNA and protein associated with inflammatory markers in MIA-injected rat model. (**A**) The mRNA expression of TNF-α, IL-1β, and IL-6; (**B**) Western blot analysis of inflammation-related protein expression; (**C**) quantification of phospho-IκBα/IκBα, phospho-p65/p65, COX-2/β-actin, and iNOS/β-actin. NC, AIN-93G; C, AIN-93G + MIA injection; PC, ibuprofen 20 mg/kg b.w. + MIA injection; CS 62, CS 62 mg/kg b.w. + MIA injection; CS 124, CS 124 mg/kg b.w. + MIA injection; CS 248, CS 248 mg/kg b.w. + MIA injection. The results are expressed as mean ± SD values (*n* = 8). Duncan’s test is used, and the different superscript letters (alphabet) indicate significance at *p* < 0.05.

**Figure 8 ijms-26-00521-f008:**
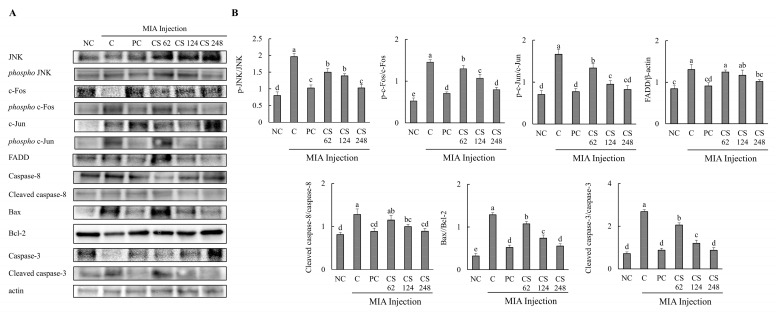
Effect of porcine-derived chondroitin sulfate sodium (CS) on the expression of protein associated with apoptosis mechanism in MIA-injected rat model. (**A**) Western blot analysis of apoptosis-related protein expression. (**B**) Quantification of phospho-JNK/JNK, phospho-c-Fos/c-Fos, phospho-c-Jun/c-Jun, FADD/β-actin, cleaved caspase-8/caspase-8, Bax/Bcl-2, cleaved caspase-3/caspase-3 expression. NC, AIN-93G; C, AIN-93G + MIA injection; PC, Ibuprofen 20 mg/kg b.w. + MIA injection; CS 62, CS 62 mg/kg b.w. + MIA injection; CS 124, CS 124 mg/kg b.w. + MIA injection; CS 248, CS 248 mg/kg b.w. + MIA injection. The results are expressed as mean ± SD values (*n* = 8). Duncan’s test is used, and the different superscript letters (alphabet) indicate significance at *p* < 0.05.

**Table 1 ijms-26-00521-t001:** Effect of CS on body weight, FER, and organ weight of MIA-induced osteoarthritis rat model.

Measurements		Induced Arthritis				
	NC	C	PC	CS 62	CS 124	CS 248
Initial body weight (g)	208.05 ± 6.45 ^NS^	212.20 ± 3.39	212.30 ± 6.09	206.83 ± 6.05	211.00 ± 5.93	212.25 ± 7.10
Final body weight (g)	352.70 ± 20.86 ^NS^	339.50 ± 13.10	353.20 ± 13.49	337.00 ± 24.16	348.00 ± 12.12	345.00 ± 26.95
Weight gain (g) ^(1)^	144.65 ± 15.66 ^NS^	127.30 ± 14.17	140.90 ± 12.85	130.17 ± 18.74	137.00 ± 8.20	132.75 ± 23.74
Food consumption (g/day)	17.98 ± 2.21 ^NS^	16.54 ± 2.09	17.91 ± 2.28	16.60 ± 2.87	17.33 ± 2.32	17.82 ± 2.44
FER ^(2)^	28.73 ± 3.11 ^NS^	27.49 ± 3.06	30.32 ± 2.77	28.01 ± 4.03	28.24 ± 1.69	26.60 ± 4.76
Liver	14.07 ± 1.89 ^NS^	13.86 ± 0.99	13.89 ± 1.79	12.59 ± 0.97	13.04 ± 0.95	12.48 ± 2.22
Kidney	2.40 ± 0.13 ^NS^	2.32 ± 0.19	2.40 ± 0.15	2.38 ± 0.15	2.43 ± 0.14	2.32 ± 0.17
Spleen	0.89 ± 0.11 ^NS^	0.84 ± 0.08	0.86 ± 0.12	0.82 ± 0.09	0.79 ± 0.11	0.78 ± 0.07

NS, Not significant. Values are presented as the mean ± standard deviation (*n* = 8), and the different superscript letters indicate significance at *p* < 0.05. NC, AIN-93G; C, AIN-93G + MIA injection; PC, ibuprofen 20 mg/kg b.w. + MIA injection; CS 62, chondroitin sulfate sodium 62 mg/kg b.w. + MIA injection, CS 124: chondroitin sulfate sodium 124 mg/kg b.w. + MIA injection; CS 248, chondroitin sulfate sodium 248 mg/kg b.w. + MIA injection. ^(1)^ Weight gain (g/4 weeks) = final body weight (g) − initial body weight (g). ^(2)^ FER (food efficiency rate) = {weight (g)/food intake (g)} × 100.

## Data Availability

The original contributions presented in this study are included in the article. Further inquiries can be directed to the corresponding authors.

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
