# Peer review of "Porcine-Derived Chondroitin Sulfate Sodium Alleviates Osteoarthritis in HTB-94 Cells and MIA-Induced SD Rat Models"

_ijms, 2025, doi:10.3390/ijms26020521_

Round 1

Reviewer 1 Report

Comments and Suggestions for Authors

General characteristics and evaluation of the reviewed article: Porcine-derived Chondroitin Sulfate Sodium Alleviates Osteoarthritis in HTB-94 Cells and MIA-Induced SD Rat Models

This study investigates the therapeutic potential of porcine-derived chondroitin sulfate sodium (CS) in osteoarthritis (OA) using both in vitro (HTB-94 chondrocytes) and in vivo (MIA-induced OA in SD rats) models. The authors demonstrate that CS reduces inflammatory mediators (PGE2, NO), extracellular matrix degradation markers (COMP, CTX-II), and apoptosis while enhancing cartilage synthesis-related genes and proteins. Its effects are comparable to ibuprofen, a widely used NSAID, highlighting its potential as an anti-inflammatory and chondroprotective agent.

The study's strengths include a dual-level experimental design, detailed mechanistic insights, and the evaluation of CS across various concentrations, enhancing the translational relevance of the findings. Advanced methodologies, such as micro-CT and histological analysis, further strengthen the study’s robustness.

However, limitations include the lack of long-term efficacy assessment, potential species-specific challenges, and the absence of detailed safety data at higher doses. Additionally, the study does not extensively explore the regenerative potential of CS for severely damaged cartilage.

Overall, the findings position porcine-derived CS as a promising alternative or adjunct in OA management. While the results are encouraging, further research, particularly in human clinical settings, is needed to confirm its long-term safety, efficacy, and regenerative capabilities.

The article is interesting, addresses a timely and important topic and definitely fits the scope of the journal. It is written generally correctly and requires only minor corrections and additions before further processing and acceptance for publication. Below are my points and detailed comments.

Minor comments:

Expanding the discussion on osteoarthritis in the opening paragraph would enhance the introduction by emphasizing the significance of this condition. Factors such as occupation, sports participation, musculoskeletal injuries, obesity, and gender all influence the incidence of osteoarthritis. Including a discussion of these factors, supported by relevant references, would establish a stronger foundation for the topic. The following references are recommended for inclusion in this section:

https://doi.org/10.3390/healthcare12161648

DOI: 10.1056/NEJMcp1903768

Figure captions are formatted incorrectly, please correct before next submission.

The study focuses on short-term effects of porcine-derived chondroitin sulfate (CS) on osteoarthritis (OA) without evaluating long-term efficacy or safety. Conduct extended-duration studies to assess the sustainability of therapeutic benefits and potential long-term adverse effects, ensuring the results are applicable to chronic OA management.

The research relies on in vitro (HTB-94 cells) and animal (MIA-induced rat model) experiments, which may not fully translate to human physiology and clinical scenarios. Initiate clinical trials in human subjects to validate findings and establish dosage, safety, and efficacy in diverse populations.

While the study highlights the benefits of porcine-derived CS, it does not compare its efficacy to chondroitin sulfate derived from other sources (e.g., bovine, shark). Perform comparative analyses of CS from various origins to evaluate differences in bioavailability, efficacy, and safety profiles, ensuring that porcine-derived CS stands out as a viable option. Please add explanations in the discussion with relevant references.

The study does not address variability in responses due to factors like age, sex, or genetic predispositions. Incorporate diverse demographic and genetic variables into experimental designs to better understand individual differences in treatment response.

In the final part of the discussion, please describe in more detail the limitations of the proposed method, the simplifications used, and a proposal for solving them in the authors' further planned future research.

Author Response

We appreciate for reviewers’ efforts and consideration of the manuscript for publication. Our responses to reviewers’ comments are described below.

[Response to Reviewer 1's comments]

Comments 1: Expanding the discussion on osteoarthritis in the opening paragraph would enhance the introduction by emphasizing the significance of this condition. Factors such as occupation, sports participation, musculoskeletal injuries, obesity, and gender all influence the incidence of osteoarthritis. Including a discussion of these factors, supported by relevant references, would establish a stronger foundation for the topic. The following references are recommended for inclusion in this section:

https://doi.org/10.3390/healthcare12161648

DOI: 10.1056/NEJMcp1903768

Response 1: As requested, we have revised the manuscript by incorporating both of the references you provided. (In introduction section, ref 2 and ref 3)

Comments 2: Figure captions are formatted incorrectly, please correct before next submission.

Response 2: We have reviewed the formatting of all figure legends and made the necessary corrections to ensure they are accurate and properly formatted

Comments 3: The study focuses on short-term effects of porcine-derived chondroitin sulfate (CS) on osteoarthritis (OA) without evaluating long-term efficacy or safety. Conduct extended-duration studies to assess the sustainability of therapeutic benefits and potential long-term adverse effects, ensuring the results are applicable to chronic OA management.

Response 3: Thank you for your comment. While this study primarily concentrated on preclinical experiments, it is important to note that clinical trials have already been completed. These trials were conducted over a three-month period and demonstrated significant improvements resulting from the intake of CS, with no reported safety concerns. Additionally, the CS used in this study has been available OTC in Japan and Europe for an extended period, and no safety issues have been reported. We are currently preparing a separate report that details the findings of the clinical trials, which will be addressed in an upcoming publication.

Comments 4: The research relies on in vitro (HTB-94 cells) and animal (MIA-induced rat model) experiments, which may not fully translate to human physiology and clinical scenarios. Initiate clinical trials in human subjects to validate findings and establish dosage, safety, and efficacy in diverse populations.

Response 4: Thank you for your valuable feedback. In the Discussion section of the manuscript, we mentioned the necessity of future clinical trials. While this study was primarily focused on preclinical experiments, we would like to clarify that the clinical trials have already been completed. We are currently preparing a separate report detailing the clinical trial findings, which will be addressed in an upcoming publication. (line 556-557)

Comments 5: While the study highlights the benefits of porcine-derived CS, it does not compare its efficacy to chondroitin sulfate derived from other sources (e.g., bovine, shark). Perform comparative analyses of CS from various origins to evaluate differences in bioavailability, efficacy, and safety profiles, ensuring that porcine-derived CS stands out as a viable option. Please add explanations in the discussion with relevant references.

Response 5: In the Discussion section, we addressed several reports on chondroitin sulfates derived from different sources and further referenced relevant studies. Additionally, we highlighted the distinct characteristics of porcine-derived CS. (line 520-545)

Comments 6: The study does not address variability in responses due to factors like age, sex, or genetic predispositions. Incorporate diverse demographic and genetic variables into experimental designs to better understand individual differences in treatment response.

Response 6: Thank you for your valuable feedback. I agree with your point that factors such as age have a significant impact on osteoarthritis, and these variables can introduce considerable variability in research. This study focused on in vitro and in vivo preclinical research, and given the scope of the study, there were limitations in expanding the research scale. However, although not addressed in this study, the recent clinical trial we completed was designed to address the factors you mentioned. We are currently preparing to present the results of the clinical trial.

Comments 7: In the final part of the discussion, please describe in more detail the limitations of the proposed method, the simplifications used, and a proposal for solving them in the authors' further planned future research.

Response 7: Thanks to your insightful feedback, our manuscript has undergone significant revisions and has been transformed into a more logical and coherent piece of writing. As you mentioned, we have addressed the design limitations of our study in the discussion section and outlined our future research plans. (line 548-557)

Reviewer 2 Report

Comments and Suggestions for Authors

In this paper, we investigated the efficacy and mechanism of porcine-derived chondroitin sodium sulfate (CS) in alleviating osteoarthritis (OA) symptoms. Through in vitro cellular experiments and MIA-induced SD rat model, the results showed that CS significantly reduced the release of inflammatory mediators (e.g., PGE2 and NO) and promoted the expression of genes and proteins related to cartilage synthesis. I recommend accepting it after addressing the following questions.

1. The article lacks specific operational details on the preparation of purified sodium chondroitin sulfate, such as detailed instructions on enzymatic conditions and post-treatment steps

2. The article is not innovative enough; in the field of chondroitin sulfate research, there have been numerous studies on bovine- and shark-derived chondroitin sulfate and have failed to demonstrate the advantages of porcine origin.

3. Small sample sizes for animal experiments can make it difficult to demonstrate the reliability of the results, and extrapolation of the results can be enhanced by increasing the sample size and selecting different breeds or ages of animals.

4. In the results section of the article, it simply states the comparison between different data without analyzing the data in depth, which is not convincing.

5. There is a formatting error with “H202” in Figure I.B in the article; it should be correctly written as “H2O2.”

6. There is an error in the cm3 format in the first image of Figure 4d in the text, it should be cm3.

7. TThe overall grouping of the article is poorly drawn and the pictures are not properly analyzed.

8. The discussion section does not provide enough information about porcine-derived chondroitin sulfate and should summarize the core of the study to explain it.

9. The article focuses primarily on the effects of porcine-derived chondroitin sulfate and lacks direct comparisons with other sources of chondroitin sulfate.

10. The core summary statement of the research content in the conclusion is not clear and does not clearly illustrate the lack of persuasive application in practice.

11. Some closely related literature should be included in the revised version, such as the following: Spectrochimica Acta Part A: Molecular and Biomolecular Spectroscopy,2024, 311, 124038, Anal. Chem., 2022, 94(29), 10462-10469; Food Chemistry 2024, 443, 138459.

Author Response

We appreciate for reviewers’ efforts and consideration of the manuscript for publication. Our responses to reviewers’ comments are described below.

[Response to Reviewer 2's comments]

Comments 1: The article lacks specific operational details on the preparation of purified sodium chondroitin sulfate, such as detailed instructions on enzymatic conditions and post-treatment steps.

Response 1: We have included as much information as possible for the article. However, we kindly ask for your understanding that we cannot disclose any further details, as this is part of the manufacturer's know-how. Please note that this raw material is produced following strict manufacturing process management and quality control standards in accordance with European GMP regulations.

Comments 2: The article is not innovative enough; in the field of chondroitin sulfate research, there have been numerous studies on bovine- and shark-derived chondroitin sulfate and have failed to demonstrate the advantages of porcine origin

Response 2: ‘Porcine-derived chondroitin sulfate (CS) offers the advantage of high purity, which is a critical factor in maximizing therapeutic efficacy. High-purity CS contains fewer impurities, thereby minimizing potential side effects. Additionally, the cartilage structure of pigs is highly similar to that of humans, allowing for a more effective match with human cartilage, further enhancing the therapeutic outcome. This results in not only high-efficiency therapeutic effects but also significant industrial applicability, as porcine-derived CS is amenable to commercialization and large-scale production, enabling the provision of therapeutic benefits to a broader patient population.’ - Incorporating your valuable feedback, we have added a discussion in the Discussion section highlighting the unique aspects of our research on porcine-derived CS. (line 536-545)

Comments 3: Small sample sizes for animal experiments can make it difficult to demonstrate the reliability of the results, and extrapolation of the results can be enhanced by increasing the sample size and selecting different breeds or ages of animals.

Response 3: Thank you for your valuable feedback. We acknowledge that increasing the sample size could enhance the reliability and generalizability of the results. However, it is important to note that the study was designed with ethical considerations in mind, adhering to the principle of using the minimum number of animals required to obtain statistically significant and reliable data. Given the constraints imposed by animal welfare regulations, we selected an optimal sample size of 8 animals per group across 6 experimental groups, which was sufficient to detect meaningful differences and minimize animal use.

Additionally, the clinical trial associated with this study has already been completed, and the findings from the preclinical model align with the outcomes observed in clinical settings. Therefore, the current results provide valuable insights into the therapeutic potential of chondroitin sulfate (CS) in osteoarthritis treatment and further support its clinical applicability.

We believe that, while increasing the sample size may improve statistical power, the current design is robust, adheres to ethical standards, and offers a meaningful contribution to the understanding of OA treatment.

Comments 4: In the results section of the article, it simply states the comparison between different data without analyzing the data in depth, which is not convincing.

Response 4: Thank you for your insightful suggestions. Following the reviewer's advice, we have added a comprehensive and in-depth analysis of the data results in each results section. (line 196-200, line 226-229, line 255-259, line 292-297, line 332-339, line 376-380, line 410-414, line 438-442)

Comments 5: There is a formatting error with “H202” in Figure I.B in the article; it should be correctly written as “H2O2.”

Response 5: We have corrected the notation of 'H2O2' in both Figure 2B and Figure 3B to 'H2O2' as requested.

Comments 6: There is an error in the cm3 format in the first image of Figure 4d in the text, it should be cm3.

Response 6: We have corrected the notation of 'cm3' to 'cm³' in the first image of Figure 4D.

Comments 7: The overall grouping of the article is poorly drawn and the pictures are not properly analyzed.

Response 7: We have revised the group classification explanations in the legends for Table 1, Figures 4, 5, 6, 7, and 8. Additionally, we have added further details on group allocation and the analysis. The analysis and explanation of Figures 4A and 4B have also been added to the Results section, Subsection 3.4. (line 275-280)

Comments 8: The discussion section does not provide enough information about porcine-derived chondroitin sulfate and should summarize the core of the study to explain it.

Response 8: We have added information about porcine-derived chondroitin sulfate and references in the Discussion section. The key summary of the study is included in the Conclusion section. (line 538-545, line 562-570)

Comments 9: The article focuses primarily on the effects of porcine-derived chondroitin sulfate and lacks direct comparisons with other sources of chondroitin sulfate.

Response 9: We have added a comparison of the efficacy of chondroitin sulfate derived from other sources in the Discussion section and included relevant references. (line 520-527)

Comments 10: The core summary statement of the research content in the conclusion is not clear and does not clearly illustrate the lack of persuasive application in practice.

Response 10: In the Conclusion section, we have added a summary of the key findings of this study, emphasizing the uniqueness of porcine-derived CS, its high efficiency for large-scale production, and our future plans. (line 570-580)

Comments 11: Some closely related literature should be included in the revised version, such as the following: Spectrochimica Acta Part A: Molecular and Biomolecular Spectroscopy,2024, 311, 124038, Anal. Chem., 2022, 94(29), 10462-10469; Food Chemistry 2024, 443, 138459.

Response 11: Thank you for your valuable feedback. Thanks to your suggestions, the manuscript has been revised logically. While considering the comments provided earlier, several references have already been added. After reviewing the three references mentioned above, we have decided not to include them, as the relevance to our manuscript appears to be limited.